# CD4, CD8b, and Cytokines Expression Profiles in Peripheral Blood Mononuclear Cells Infected with Different Subtypes of KoRV from Koalas (*Phascolarctos cinereus*) in a Japanese Zoo

**DOI:** 10.3390/v12121415

**Published:** 2020-12-09

**Authors:** Mohammad Enamul Hoque Kayesh, Md Abul Hashem, Fumie Maetani, Taiki Eiei, Kyoya Mochizuki, Shinsaku Ochiai, Ayaka Ito, Nanao Ito, Hiroko Sakurai, Takayuki Asai, Kyoko Tsukiyama-Kohara

**Affiliations:** 1Transboundary Animal Diseases Centre, Joint Faculty of Veterinary Medicine, Kagoshima University, Kagoshima 890-0065, Japan; mehkayesh@yahoo.com (M.E.H.K.); mdhashem29@yahoo.com (M.A.H.); 2Department of Microbiology and Public Health, Faculty of Animal Science and Veterinary Medicine, Patuakhali Science and Technology University, Barishal 8210, Bangladesh; 3Department of Health, Chattogram City Corporation, Chattogram 4000, Bangladesh; 4Hirakawa Zoological Park, Kagoshima 891-0133, Japan; maetani@k-kouenkousya.jp (F.M.); eiei@k-kouenkousya.jp (T.E.); mochizuki@k-kouenkousya.jp (K.M.); ochiai@k-kouenkousya.jp (S.O.); a-itou@k-kouenkousya.jp (A.I.); itou@k-kouenkousya.jp (N.I.); sakurai@k-kouenkousya.jp (H.S.); asai@kyusyu-macaca.jp (T.A.)

**Keywords:** koala, koala retrovirus, peripheral blood mononuclear cells, CD4, CD8b, cytokines

## Abstract

Koala retrovirus (KoRV) poses a major threat to koala health and conservation, and currently has 10 identified subtypes: an endogenous subtype (KoRV-A) and nine exogenous subtypes (KoRV-B to KoRV-J). However, subtype-related variations in koala immune response to KoRV are uncharacterized. In this study, we investigated KoRV-related immunophenotypic changes in a captive koala population (Hirakawa zoo, Japan) with a range of subtype infection profiles (KoRV-A only vs. KoRV-A with KoRV-B and/or -C), based on qPCR measurements of CD4, CD8b, IL-6, IL-10 and IL-17A mRNA expression in unstimulated and concanavalin (Con)-A-stimulated peripheral blood mononuclear cells (PBMCs). Although CD4, CD8b, and IL-17A expression did not differ between KoRV subtype infection profiles, IL-6 expression was higher in koalas with exogenous infections (both KoRV-B and KoRV-C) than those with the endogenous subtype only. IL-10 expression did not significantly differ between subtype infection profiles but did show a marked increase—accompanying decreased CD4:CD8b ratio—in a koala with lymphoma and co-infected with KoRV-A and -B, thus suggesting immunosuppression. Taken together, the findings of this study provide insights into koala immune response to multiple KoRV subtypes, which can be exploited for the development of prophylactic and therapeutic interventions for this iconic marsupial species.

## 1. Introduction

The koala (*Phascolarctos cinereus*) is facing threats from habitat loss to bushfire in its native Australia [1], and the most interesting of these issues for virologists is koala retrovirus (KoRV). This virus poses a threat to both wild and captive populations, and has been implicated in neoplasia and chlamydial diseases [2,3,4,5,6,7,8]. KoRV is a gammaretrovirus in the family Retroviridae, and was first identified in 1988 in a leukemic koala [9]. Its replication-competent full genome sequence has been reported recently [10]. KoRV has a positive-sense, single-stranded RNA genome of 8.4 kb that contains gag, pol, and env genes, and long terminal repeats (LTRs) at the 5′ and 3′ ends [3,10,11]. KoRV’s closest genetic relative is gibbon ape leukemia virus (GALV) [10,12], and it is also closely related to feline leukemia virus (FeLV) and porcine endogenous retrovirus (PERV) [6]. Koalas are considered to have been first infected with this virus by transspecies transmission [12,13,14,15,16]. KoRV exists in both exogenous and endogenous forms. The KoRV endogenization process begun relatively recently, with entry into the koala genome within the last 100 years, and appears to be still ongoing, whereas other retroviruses’ endogenization into mammalian genomes occurred millions of years ago [11,14].

The endogenized form of KoRV is KoRV-A, and nine other subtypes (KoRV-B to J) have been identified to date [17]. Of these subtypes, KoRV-A and KoRV-B have been characterized to the greatest extent [18,19]. KoRV prevalence varies based on geographical location of a particular population [20]. KoRV-A has 100% prevalence in northern Australian koala populations, and KoRV-A is fully endogenized in northern koalas [8,20,21]. However, southern koalas currently have less KoRV-A prevalence with little or no evidence for endogenization [18,22,23]. KoRV-B was recently demonstrated to be an exogenous variant associated with immunosuppression, chlamydial diseases, and neoplasia [5,8,18,24], and other subtypes have also been reported.

KoRV subtypes may differ in prevalence and their endogenous or exogenous nature; however, they appear to share a sequence for the KoRV transmembrane envelope protein p15E, which contains an immunosuppressive domain and is highly conserved in different retroviruses [25,26]. This begs the question of whether koalas have a uniform or varied response to the different subtypes of the virus. KoRV has been subject to much epidemiological research, and comparative studies of its immunomodulatory effects in populations where endogenization is not yet complete. The latter studies have concentrated on evaluating differences between KoRV-positive and KoRV-negative koalas [27,28,29,30,31,32]. However, there are no studies on variations in immune response to KoRV by subtype in koala populations where multiple exogenous subtypes are found in addition to the fully endogenized subtype. A proper understanding of their immune response to different KoRV subtypes is required for successful preventive and therapeutic interventions.

Accordingly, in this study, we investigated the immune response variations in koalas to different KoRV subtypes, by measuring CD4, CD8b, and cytokines (IL-6, IL-10 and IL-17A) mRNA expression in koala peripheral blood mononuclear cells (PBMCs). We targeted three subtypes for investigation: the endogenous subtype (KoRV-A) and two exogenous subtypes (KoRV-B and KoRV-C). Additionally, we investigated the expression of immune molecules in tissues from a dead joey and a dead adult koala, to throw further light on immune responses mounted to the virus by this marsupial species.

## 2. Materials and Methods

### 2.1. Sample Collection

We targeted captive koalas (two males, eight females; age range: 6 months to 12 years) maintained in air conditioning system (23~25 °C) in a Japanese zoo (Hirakawa Zoological Park, Kagoshima, Japan) (Table 1). One adult female koala died at a later point in the study period, and her bone marrow was included in an investigation of tissue expression together with heart, lung, and spleen tissue from a joey whose death and KoRV subtype genotyping we previously reported [24]. Whole blood samples were collected by venipuncture using EDTA between January 2019 and July 2020. Where unknown, the infection status of koalas was determined for three KoRV subtypes, KoRV-A (the endogenous subtype) and KoRV-B and KoRV-C (exogenous subtypes). This study was performed in accordance with the protocols of the Institutional Animal Care and Use Committee of the Joint Faculty of Veterinary Medicine, Kagoshima University, Japan.

### 2.2. Hematological Examination

Hematological examination was performed using standard protocols to determine white blood cell (WBC), red blood cell (RBC), hemoglobin level, hematocrit, mean corpuscular volume, and mean corpuscular hemoglobin concentration.

### 2.3. Peripheral Blood Mononuclear Cells (PBMCs) Culture

Koala PBMCs were isolated from whole blood samples as described previously [33], with some modifications [34]. The isolated PBMCs were suspended in RPMI 1640 growth medium containing 20% heat-inactivated FCS, 1 mM sodium pyruvate, 50 µM β-mercaptoethanol, 1% MEM non-essential amino acids, penicillin (100 units/mL), and streptomycin (100 µg/mL). Isolated PBMCs from each individual sample were plated into two wells of a 6-well plate, one well was kept unstimulated, another well was stimulated with 20 µg of concanavalin A (Con–A) per mL, and all wells were incubated at 37 °C in a 5% CO_2_ incubator for 5 h. After incubation PBMCs were harvested, and total RNA expression was quantified in cells promptly extracted using RNeasy Mini Kit (QIAGEN, Hilden, Germany) in accordance with the manufacturer’s instructions. The concentration and purity of the extracted RNA samples was confirmed using a NanoDrop ND-1000 spectrophotometer (NanoDrop Technologies, Inc., Waltham, MA USA), and RNA samples were kept at −80 °C until use.

### 2.4. Extraction of Genomic DNA from PBMCs and Determination of KoRV Subtypes

Genomic DNA (gDNA) was extracted from EDTA-treated koala whole blood samples using Wizard Genomic DNA Purification Kit (Promega, Madison, WI, USA) following the manufacturer’s instructions. Extracted gDNA was used as a template for the PCR analysis for KoRV provirus determination, and also for KoRV subtyping. KoRV provirus was confirmed by PCR analysis, using primers targeting KoRV pol gene, pol F (5′-CCTTGGACCACCAAGAGACTTTTGA-3′) and pol R (5′-TCAAATCTTGGACTGGCCGA-3′), as described previously [11]. Genotyping PCR was then performed for the detection KoRV-A, KoRV-B, and KoRV-C using subtype-specific primers targeting KoRV envelope gene (Table 2) [5,8,24]. PCR conditions comprised denaturation at 98 °C for 2 min, 35 cycles at 98 °C for 30 s, annealing at 63 °C for 30 s, extension at 72 °C for 1 min, and extension at 72 °C for 5 min. Partial env gene PCR fragments that generated from subtype-specific PCR were subcloned into pCR-Blunt II TOPO (Invitrogen) and sequenced. The resulting sequences were submitted to the GenBank database (GenBank accession nos. MT951447–MT951453).

### 2.5. Extraction of RNA from Koala Tissues

Tissues (spleen, lung, and heart, or bone marrow) previously collected from a dead, six-month-old joey [24] were submitted for further analysis in this study. Bone marrow was collected from an adult koala (KM) with lymphoma that died during the study period and submitted for expression analysis (Table 1). Tissue samples were stored at −80 °C until extraction of RNA. Total RNA was extracted from tissues using RNeasy Plus Mini Kit (QIAGEN), according to the manufacturer’s instructions. The concentration and purity of the extracted RNA were determined as described above. RNA samples were stored at −80 °C until use.

### 2.6. Gene Expression Analysis by Quantitative Reverse Transcription-PCR (qRT-PCR)

CD4, CD8b, IL-6, IL-10, and IL-17A expression levels were quantified in both unstimulated or ConA-stimulated PBMCs from koalas, and tissue samples from dead koalas, using one-step qRT-PCR with Brilliant III Ultra-Fast SYBR Green qRT-PCR Master Mix (Agilent Technologies, Santa Clara, CA, USA) according to the manufacturer’s instructions. Each reaction was performed in duplicate in a 20 μL volume in a 96 micro-well plate using the CFX Connect Real-Time PCR Detection System (Bio-Rad). The cycling conditions were as follows: reverse transcription at 50 °C for 10 min, initial denaturation at 95 °C for 3 min, and 40 cycles of 95 °C for 5 s and 60 °C for 10 s. Specificity of the PCR reaction was confirmed by melt curve analysis. Each reaction included a no-template control and a standard curve for each gene. Standards were prepared from the prequantified plasmids containing the target gene sequence. Primers used for amplification of the target and reference genes that were taken from previous studies or designed using Primer-Blast (https://www.ncbi.nlm.nih.gov/tools/primer-blast/) based on target sequences available in GenBank are shown in Table 2. Koala beta actin was used as an endogenous control for normalization of the results.

### 2.7. Statistical Analysis

Analyses were performed to compare infection subtype profiles, based on positive or negative results for KoRV-A, KoRV-B, and KoRV-C. Data are presented as the mean ± standard deviation (SD). A Student’s t-test was performed using GraphPad software for statistical analysis. *p*-values < 0.05 were considered statistically significant.

## 3. Results

### 3.1. KoRV Infection Status of Koalas

All nine koalas targeted for hematological investigation of KoRV subtype-related variations in immunological parameters (CD4, CD8b, and cytokine expression) were positive for KoRV provirus in PCR testing with KoRV pol gene-specific primers on genomic DNA extracted from their PBMCs.

To determine the infection status of each animal by KoRV subtype, samples were subject to genotyping PCR and sequencing, and the results are shown in Table 1. Infection status had been determined for four koalas (H6, H7, H8, and H9) in previous studies [24,35]. All koalas were positive for the endogenous subtype, KoRV-A. The status for exogenous infection was as follows 4/9 koalas were negative for both KoRV-B and KoRV-C (Koalas KY, KYB, H6, and KJ); 3/9 koalas were positive for both KoRV-B and KoRV-C (Koalas KS, H7, and H9); 1/9 koalas was positive for KoRV-B but negative for KoRV-C (KM); and 1/9 koalas was positive for subtype KoRV-C but negative for KoRV-B (H8).

### 3.2. Hematological Examination of Koalas

The hematology results for the koalas evaluated in this study are shown in Table 3. Blood parameters were within normal ranges for seven of the nine koalas (KY, KYB, H6, H7, H8, H9, and KJ), which were designated as “apparently healthy”. The other two koalas (KS and KM) were found to have lymphoma.

### 3.3. Changes of CD4, CD8b, and Cytokines mRNA Expression in Unstimulated Koala PBMCs

To evaluate variations in immune response according to KoRV subtype infection status, we measured expression levels for CD4, CD8b, IL-6, IL-10, and IL-17A in RNA isolated from unstimulated koala PBMCs. CD4 and CD8b showed no significant variation between infection subtype profiles in mRNA expression in unstimulated koala PBMCs (Figure 1A,B). IL-6 mRNA expression was significantly higher in koalas positive for both exogenous subtypes (KoRV-B and KoRV-C) than in those with endogenous infection only (KoRV-A; H6, KJ, KY, and KYB) (Figure 1C). IL-10 mRNA expression was markedly higher in the KoRV-B-positive, KoRV-C-negative individual (KM, which showed lymphoma) than in koalas with endogenous infection only (KoRV-A) (Figure 1D). IL-17A showed no significant variation between infection subtype profiles in mRNA expression (Figure 1E).

### 3.4. Fold Change of CD4, CD8b, and Cytokines Expression in Stimulated Koala PBMCs

To evaluate potentially enhanced expression after stimulation, we determined CD4, CD8b, IL-6, IL-10, and IL-17A expression levels in Con-A stimulated koala PBMCs relative to those in unstimulated koala PBMCs, and found no significant differences between subtype profiles (Figure 2A–E).

### 3.5. Changes in CD4 and CD8b Ratio

To further evaluate changes following stimulation, we measured the CD4:CD8b ratio in unstimulated koala PBMCs, and evaluated the corresponding ratio in Con-A stimulated koala PBMCs for relative change. CD4:CD8b ratio in unstimulated koala PBMCs was markedly higher in the KoRV-B-positive, KoRV-C-negative individual (KM, which showed lymphoma) than in koalas with endogenous infection only (KoRV-A); however, this ratio showed no other significant differences between KoRV subtype infection profiles (Figure 3A). After Con-A stimulation, the fold change in CD4:CD8b ratio (vs. unstimulated PBMCs) was markedly increased in one koala with endogenous infection only (H6; KoRV-A positive) and one koala positive for both exogenous subtypes (H7; KoRV-A, -B, and -C positive) (Figure 3B).

### 3.6. Expression Pattern of CD4, CD8b, and Cytokines mRNA in Koala Tissues

To investigate immunological parameters in koala tissues, we quantified CD4, CD8b, IL-6, IL-10, and IL-17A mRNA expression in the spleen, lung and heart tissues of a dead joey whose infection status (KoRV-A and KoRV-C positive) and tissue proviral loads were reported in our previous study [24], and the bone marrow of the adult koala (KM) with lymphoma that died during the study period. The joey showed differential expression of immune molecules in spleen, lung, and heart tissues (Figure 4A–C), with IL-17A highly expressed in spleen and lung, but IL-6 highly expressed in heart (Figure 4A–C). The adult koala that died showed high IL-6 expression, but IL-17A was undetectable (Figure 4D). We determined CD4:CD8b ratio in these tissues. In the joey, CD4:CD8b ratio was higher in the lung than the spleen or heart. The adult koala that died (KM) showed a suppressed CD4:CD8b ratio with a value below one (0.72) (Figure 4E) indicating immunosuppression.

## 4. Discussion

To the authors’ knowledge, this is the first study to compare CD4, CD8b, and cytokines response between koalas with only endogenous KoRV infection, and koalas additionally infected with exogenous subtypes (KoRV-B and/or KoRV-C). We set out to ascertain any KoRV subtype-dependent immunological variations by characterizing CD4, CD8b, IL-6, IL-10, and IL-17A mRNA expression in koala PBMCs. Furthermore, we characterized the expression levels of these immune molecules in tissues from an adult koala and a joey to gain an insight into relative immune responses.

As a novel finding in this study, koalas with two exogenous KoRV subtypes (KoRV-B and KoRV-C) in addition to the endogenous KoRV subtype (KoRV-A) showed significantly higher expression of IL-6 in PBMCs than koalas with only the endogenous infection. This is consistent with the previously reported increases in IL-6 and IL-10 in human PBMCs incubated with KoRV [36], and these findings are indicative of inflammation and immunosuppression.

Our study presents a contrast to another recent study in free-ranging koalas. That study of a wild population in southwestern Australia (Victoria)—where KoRV has yet to be fully endogenized into the genome—demonstrated that KoRV-positive koalas had a significantly lower IL-17A level than KoRV-negative koalas; furthermore, the only KoRV subtype present in that study population was KoRV-A [32]. The population in our study reflected the situation in northern Australia, where KoRV-A has been fully endogenized, and the comparison in our study was between different infection subtype profiles, koalas with the endogenous infection (KoRV-A) only vs. those with either or both KoRV-B and KoRV-C, which are exogenous forms of the virus. Interestingly, we found no differences in IL-17A expression between koalas with only the endogenous infection and those who also had an exogenous infection (KoRV-B and/or KoRV-C).

We also made interesting findings with regard to CD4:CD8 ratios, which can be utilized as a potential biomarker for immunological evaluation [37,38,39,40]. The above-mentioned study in a wild population in southwestern Australia revealed a significantly lower CD4:CD8 ratio in KoRV-positive (KoRV-A) koalas, than KoRV-negative koalas [32]. In this study, the CD4:CD8b ratio in unstimulated koala PBMCs did not differ significantly with infection subtype profile (endogenous only vs. endogenous plus one or two exogenous subtypes). However, as might be expected based on previous research in koalas [4,5], the KoRV-A- and B-positive koala with lymphoma that died in this study showed a markedly decreased CD4:CD8b ratio and markedly increased IL-10 expression.

An altered CD4:CD8 ratio accompanied with differences in cytokine expression is already known to be associated with KoRV-B infection and this ratio is reportedly subject to seasonal variation in captive koalas [29]. KoRV contains transmembrane protein p15E5, which is highly conserved among different KoRV subtypes, and linked to immune suppressive effects including inhibition of lymphocyte activation by mitogens and modulation of cytokine expression by PBMCs [6,26]. Similarly, stimulation of koala PBMCs in this study yielded no significant differences in relative expression level for CD4, Cd8b, or cytokines.

The expression of immune molecules was investigated in tissue from a dead joey with the endogenous infection only (KoRV-A) and an adult, KoRV-A- and B-positive koala that died in the study period (and had been found to have lymphoma). The joey showed differential expression of CD4, CD8b, IL-6, IL-10, and IL-17A (in the spleen, heart, and lung). The exogenously infected adult koala that died showed high IL-6 expression with undetectable levels of IL-17A, findings suggestive of inflammation and immunosuppression, respectively, in bone marrow. Furthermore, the joey, which was negative for exogenous KoRV-B infection, showed CD4:CD8b ratios above one in all tissues examined, whereas the subtype B- and C-positive showed a CD4:CD8b ratio below one in the tissue examined (bone marrow), a finding indicative of immunosuppression [18].

## 5. Conclusions

In conclusion, we evaluated differences in PBMC immune response between koalas with only an endogenous infection (KoRV-A) and those with exogenous infections (KoRV-B and/or KoRV-C). This study population reflected the situation seen in the wild in northern Australia, where KoRV-A has been fully endogenized, and there are no KoRV-negative koalas. We found increased IL-6 expression in koalas with exogenous infections (both KoRV-B and KoRV-C), findings indicative of inflammation. Further investigation of a KoRV-B- and C-positive adult koala that died (found to have lymphoma) revealed immunosuppression characterized by a decreased CD4:CD8b ratio and increased IL-10 expression. Future investigations require larger koala populations, expression analyses in a wider range of tissues, and expanding the targeted KoRV subtypes, to enhance understanding of immune response mechanisms and facilitate progress on prophylactic and therapeutic interventions strategies for this iconic marsupial species.

## Figures and Tables

**Figure 1 viruses-12-01415-f001:**
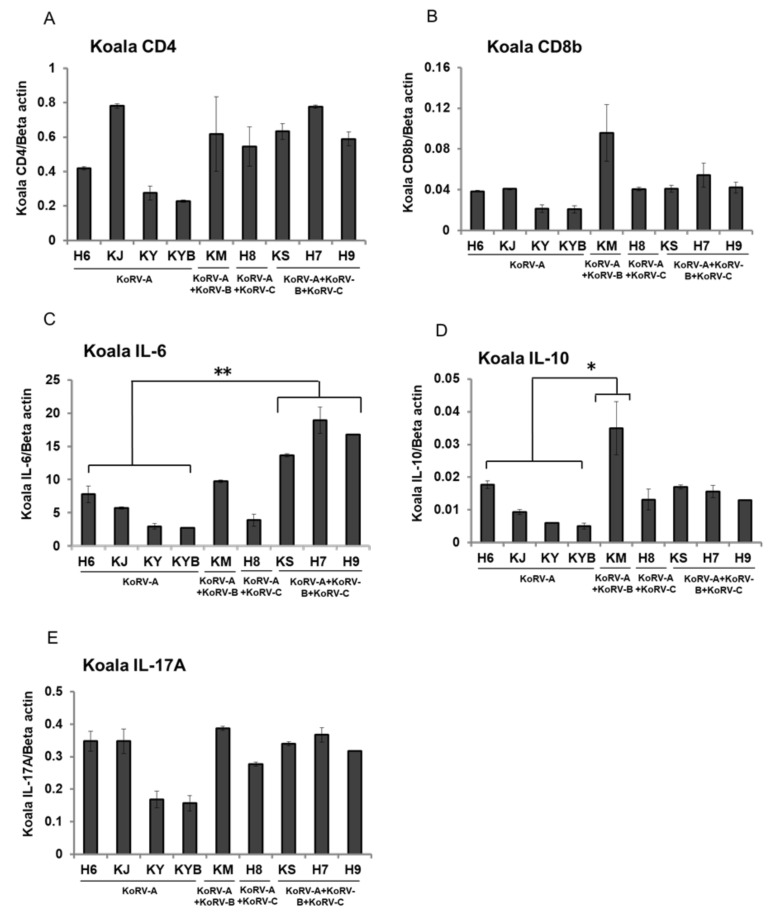
CD4, CD8b, and cytokines mRNA expression level in unstimulated koala peripheral blood mononuclear cells (PBMCs). mRNA expressions of CD4 (**A**), CD8b (**B**), IL-6 (**C**), IL-10 (**D**), and IL-17A (**E**) are indicated in unstimulated koala PBMCs infected with different KoRV subtypes. The transcript levels were normalized against koala beta actin. Statistical significance was calculated using the Student’s *t*-test and indicated by asterisks (* *p* <  0.05, ** *p* < 0.01). Data are presented as mean ± SD (*n* = 2). KoRV-A positive only koalas were considered as the control.

**Figure 2 viruses-12-01415-f002:**
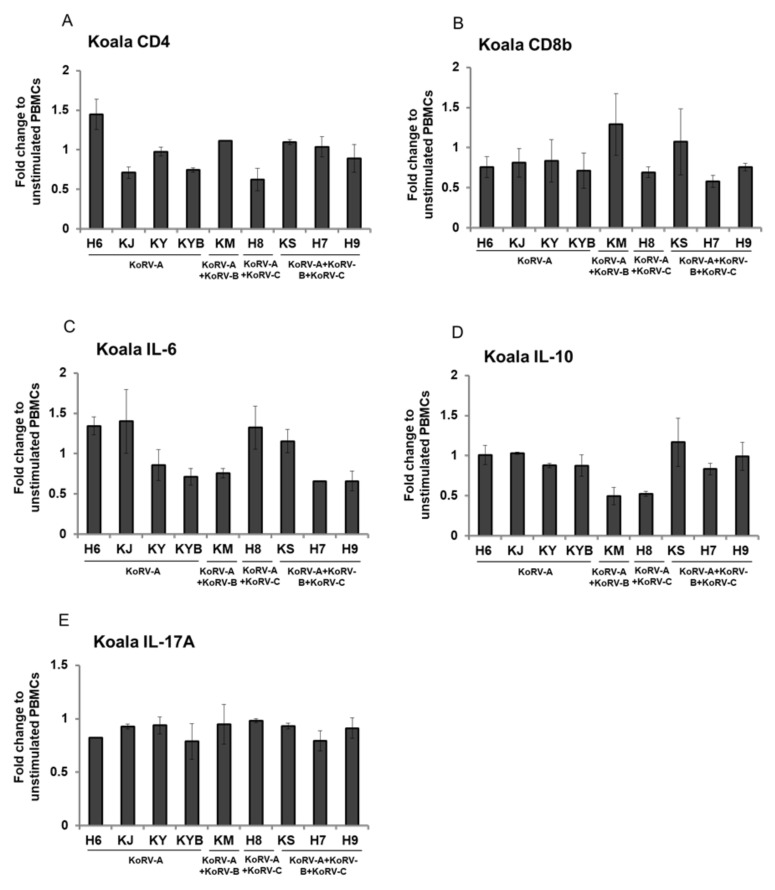
Expression of CD4, CD8b, and cytokines in Con-A-stimulated koala PBMCs. mRNA expressions of CD4 (**A**), CD8b (**B**), IL-6 (**C**), IL-10 (**D**), and IL-17A (**E**) in koala PBMCs infected with KoRV-B and/or KoRV-C along with KoRV-A are shown. RNA levels were normalized with koala beta actin, and the ratio to unstimulated koala PBMCs is indicated. Data are presented as mean ± SD (*n* = 2).

**Figure 3 viruses-12-01415-f003:**
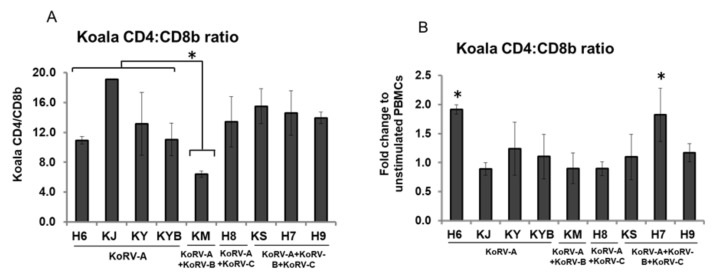
CD4:CD8b ratio. CD4:CD8b ratio of mRNA expression in unstimulated koala PBMCs (**A**). CD4:CD8b ratio in Con-A-stimulated koala PBMCs to unstimulated koala PBMCs is indicated (**B**). KoRV-A positive only koalas were considered as the control. Statistical significance was calculated using the Student’s t-test and indicated by asterisks (* *p* < 0.05). Data are presented as mean ± SD (*n* = 2).

**Figure 4 viruses-12-01415-f004:**
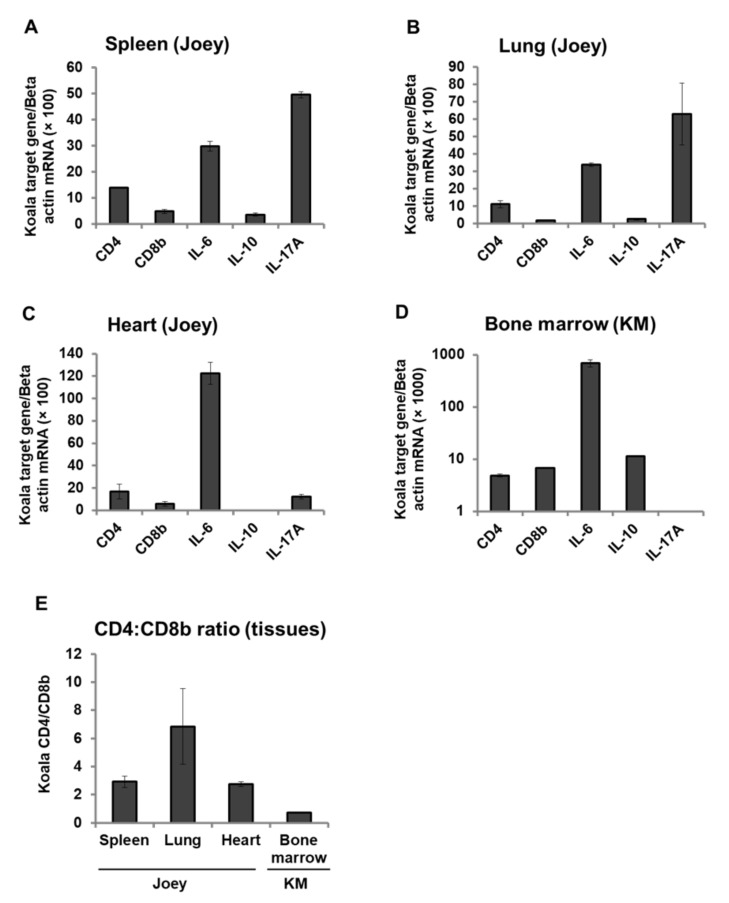
CD4, CD8b, and cytokines mRNA expression in koala tissues. Expression pattern of CD4, CD8b, IL-6, IL-10, and IL-17A mRNA in spleen (**A**), lung (**B**), heart (**C**) of a dead joey, and bone marrow of KM (**D**) is shown. The transcript levels were normalized against koala beta actin. CD4:CD8 ratio of mRNA expression in koala tissues from joey and KM (**E**) is shown.

**Table 1 viruses-12-01415-t001:** Koalas used in this study.

Koala	Age	Sex	KoRV Subtypes			Health Status
			KoRV-A	KoRV-B	KoRV-C	
KS	1 y 6 m	Male	Positive	Positive	Positive	Lymphoma
KM	12 y	Female	Positive	Positive	Negative	Lymphoma (subsequently died)
KY	2 y	Female	Positive	Negative	Negative	Apparently healthy
KYB	1 y	Female	Positive	Negative	Negative	Apparently healthy
H6	3 y 9 m	Female	Positive	Negative	Negative	Apparently healthy
H7	3 y 1 m	Female	Positive	Positive	Positive	Apparently healthy
H8	5 y 5 m	Female	Positive	Negative	Positive	Apparently healthy
H9	10 y 3 m	Female	Positive	Positive	Positive	Apparently healthy
KJ	6 y	Female	Positive	Negative	Negative	Apparently healthy
Joey	6 m	Male	Positive	Negative	Positive	Dead

**Table 2 viruses-12-01415-t002:** Primers used for the detection of koala retrovirus (KoRV) (subtypes) and gene expression analysis.

Gene	Forward (5′ to 3′)	Reverse (5′ to 3′)	Product Size	Reference or GenBank Accession Numbers
Pol (KoRV)	CCTTGGACCACCAAGAGACTTTTGA	TCAAATCTTGGACTGGCCGA	523	[11]
Env (KoRV-A)	TCCTGGGAACTGGAAAAGAC	GGGTTCCCCAAGTGATCTG	321	[8]
Env (KoRV-B)	TCCTGGGAACTGGAAAAGAC	GGCGCAGACTGTTGAGATTC	271	[5]
Env (KoRV-C)	TCCTGGGAACTGGAAAAGAC	AAGGCTGGTCCCGCGAAGGT	290	[24]
CD4 (Koala)	GCCAACCCAAGTGACTCTGT	TCTCCTGGACCACTCCATTC	105	[27]
CD8b (Koala)	GCATTGGCTTCTAATTGCTAGTATC	CACTTTCTATCATGCAAAGTAACCC	88	XM_020969485
IL-6 (Koala)	TGGATGAGCTGAACTGTACCC	GCTTGCCAAGGATTGTGAGT	119	[27]
IL-10 (Koala)	ACCAGAGACAAGCTCGAAAC	TCTTCCAGCAAAGATTTGTCTATC	50	XM_021002936
IL-17A (Koala)	GAGGCTAGGTGCCGTCATTC	TGCTGGATTTCGACGGAGTT	80	KJ174517
Beta actin (Koala)	AGATCATTGCCCCACCT	TGGAAGGCCCAGATTC	123	[11]

**Table 3 viruses-12-01415-t003:** Hematological data from koalas.

Koala	Blood Parameters						
	WBC (10^2^/μL)	RBC (10^4^/μL)	HGB (g/dL)	PCV (%)	MCV (fL)	MCH (pg)	MCHC (g/dL)
KS	4000	23.4	8.8	26	111.1	37.6	33.8
KM	2275	241	8.8	25.4	105.4	36.5	34.6
KY	102	360	14.1	40.5	112.5	39.2	34.8
KYB	52	308	12	35.9	116.6	39	33.4
H6	70	390	14	41.6	106.7	35.9	33.7
H7	99	349	13.7	39.1	112	39.3	35
H8	82	342	13	38	111.1	38	34.2
H9	52	319	11	33.8	106	34.5	32.5
KJ	125	359	13.2	37.8	105.3	36.8	34.9

WBC, white blood cell; RBC, red blood cell, HGB, hemoglobin; PCV, packed cell volume; MCV, mean corpuscular volume; MCH, mean corpuscular hemoglobin; MCHC, mean corpuscular hemoglobin concentration. Blood parameter data for Koalas H6, H7, H8, and H9 are from our recent study [35] and are shown as a reference.

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
