# Peer review of "CD4, CD8b, and Cytokines Expression Profiles in Peripheral Blood Mononuclear Cells Infected with Different Subtypes of KoRV from Koalas (Phascolarctos cinereus) in a Japanese Zoo"

_viruses, 2020, doi:10.3390/v12121415_

Round 1
Reviewer 1 Report
It would have been interesting if the authors had indicated in their conclusions what their results imply in terms of the health of individual, free-ranging koalas or their populations in the wild, given the differences in prevalence of the virus in different geographic areas of Australia.
Lines 73-73. The authors state, "A proer understanding of their imune response to different KoRV subtypes is crucial fo successful preventive and theraputic interventions." That might be true for captive koalas but is it an overstatement for koalas in wild populations? What might feasible measures be for wild populations?
Author Response
Response: We are grateful to the reviewer for his sincere comments.
Comments and Suggestions for Authors
It would have been interesting if the authors had indicated in their conclusions what their results imply in terms of the health of individual, free-ranging koalas or their populations in the wild, given the differences in prevalence of the virus in different geographic areas of Australia.
Response: Thanks for the reviewer comments. As in this study we could use only limited number of captive koalas future investigations with larger koala populations are required for better implication of the findings of this study. However, increased IL6 expression in koalas infected with KoRV-B and KoRV-C subtypes was indicated in the conclusion (line 314-315), which might be reflected in the wild population.
Lines 73-73. The authors state, "A proer understanding of their imune response to different KoRV subtypes is crucial fo successful preventive and theraputic interventions." That might be true for captive koalas but is it an overstatement for koalas in wild populations? What might feasible measures be for wild populations?
Response: Thanks for the reviewer comments. In response to reviewer comments we have rephrased the sentence (line 73). Although difficult, however compared to other strategies vaccination (of which development is underway now) might be feasible for wild populations.
Reviewer 2 Report
The only published report establishing KoRV-C as a KoRV subtype is from a Japanese Zoo (Shojima T, Yoshikawa R, Hoshino S, Shimode S, Nakagawa S, Ohata T, Nakaoka R, Miyazawa T. 2013. Identification of a novel subgroup of Koala retrovirus from Koalas in Japanese zoos. J Virol 87:9943-8). The present study is thus really interesting as it is also the first to investigate the immunological effect of KoRV-C in koalas.
Line 56-59: While I understand the authors' intentions here, the reader may wrongly infer that 100% infection rate always means endogenization. You may want to restructure this section to better reflect the facts that (a) KoRV-A has 100% prevalence in northern Australia koala populations, (b) KoRV-A is fully endogenized in northern koalas, (c) and finally, southern koalas currently have less KoRV-A prevalence with little or no evidence for endogenization.
Line 75: The authors claimed to investigate the immune response to different KoRV subtypes. This is untrue as the PBMCs used in this study were either left unstimulated or stimulated with Con-A. To claim that the immune response measured was directed against KoRV, you would have to stimulate the PBMCs with either KoRV (or its recombinant protein or peptides). Given how hard it may be to obtain pure KoRV cultures, the authors might want to soften the claim of measuring immune response directed against different KoRV subtypes.
Line 88: The authors stated that sampling was done between January 2019 and July 2020. A recent study, also cited by the authors (Maher et al., 2016), showed an association between koala immune cytokine expression and season. Did the authors take sampling season into account, especially when measuring blood parameters?
Line 168 (Table 3): It will be highly beneficial to the reader if the authors can add the Health status column in Table 3 to Table 1 as it helps the reader understand better the possible contribution of KoRV subtypes to lymphoma.
Line 196: Since these bars in the figure are for individual koalas, kindly explain the error bars shown. Do they refer to the standard deviation of replicates of PBMCs culture (stimulated or unstimulated) or PCR duplicates? Explain this and how many replicates (if PBMC) either in the figure legends or methods section. The same should be done for all figures 1-3.
Line 219: what do the dashed horizontal lines in the figure indicate? Also, it might be better to present the fold change graphs as a Log2 fold change as such graphs are nicer because they centre around zero. For instance, a fold change of 0.5 would be shown as -1 while a fold change of 2 will be shown as 1 (i.e. fold change = 0.5, Log2FC = -1; fold change=2, Log2FC = 1). Just a suggestion.
Also, for all the bar charts, can the authors attempt to overlay a scatter/dot plot on the bars so the reader can appreciate the individual points being plotted?
Line 227: The authors reported a lower CD4:CD8b ratio in one koala (KM) relative to the other koalas. Did the authors account for seasonal differences as previously reported by Maher et al., 2016 (ref #29)? Was any other koala sampled in similar season? If yes, all good but if not, it might be hard to state that the lowered CD4:CD8 ratio is not a result of seasonal changes in koala immune profile.
Line 267: A previous study also tested immune response in KoRV-A vs KoRVA+B positive koalas in terms of their antibody response to the KoRV antigen (Olagoke et al., 2019; ref #31). I understand that that study did not investigate cytokine or CD4/8 response, but antibody response is also a form of immune response. As such, can the authors rephrase the statement to emphasise the type of immune response they have studied?
Line 278-287: Were these koalas ancestors sourced from Queensland? If yes, that might explain why they behave similarly to northern Australia koala populations re IL17 levels.
Line 288: while it is true that the CD4:CD8 ratio can be used as a biomarker for immune evaluation, the truth is that the CD4:CD8 ratio traditionally refers to CD4 and CD8 positive T lymphocytes and which are typically quantified at the cellular and not mRNA level by flow cytometry. To my knowledge, there is no study published study yet on how CD4 and CD8 mRNA levels correlate with actual T lymphocytes counts in koalas. I would argue that you include this point in your interpretation of your really interesting data.
The manuscript may benefit from an alignment of the KoRV-A, B and C transmembrane protein p15E to highlight why koalas with exogenous KoRV koalas might be more at risk of immunosuppression. This is just a suggestion and the authors are free to ignore if uninterested.
Author Response
Response: We would like to thank the reviewer for his sincere comments.
Comments and Suggestions for Authors
The only published report establishing KoRV-C as a KoRV subtype is from a Japanese Zoo (Shojima T, Yoshikawa R, Hoshino S, Shimode S, Nakagawa S, Ohata T, Nakaoka R, Miyazawa T. 2013. Identification of a novel subgroup of Koala retrovirus from Koalas in Japanese zoos. J Virol 87:9943-8). The present study is thus really interesting as it is also the first to investigate the immunological effect of KoRV-C in koalas.
Response: We are very grateful to the reviewer for his sincere comments.
Line 56-59: While I understand the authors' intentions here, the reader may wrongly infer that 100% infection rate always means endogenization. You may want to restructure this section to better reflect the facts that (a) KoRV-A has 100% prevalence in northern Australia koala populations, (b) KoRV-A is fully endogenized in northern koalas, (c) and finally, southern koalas currently have less KoRV-A prevalence with little or no evidence for endogenization.
Response: Thanks for the reviewer comments. In line up with the reviewer comments we have updated the text (line 56-64).
Line 75: The authors claimed to investigate the immune response to different KoRV subtypes. This is untrue as the PBMCs used in this study were either left unstimulated or stimulated with Con-A. To claim that the immune response measured was directed against KoRV, you would have to stimulate the PBMCs with either KoRV (or its recombinant protein or peptides). Given how hard it may be to obtain pure KoRV cultures, the authors might want to soften the claim of measuring immune response directed against different KoRV subtypes.
Response: Thanks for the reviewer comments. Agreeing with reviewer comments we have modified the text to soften the claim of measuring immune response directed against different KoRV subtypes (line 78).
Line 88: The authors stated that sampling was done between January 2019 and July 2020. A recent study, also cited by the authors (Maher et al., 2016), showed an association between koala immune cytokine expression and season. Did the authors take sampling season into account, especially when measuring blood parameters?
Response: We thank the reviewer for his nice comments. In our study we used captive koalas reared in a Japanese zoo, where the koalas are housed in air conditioning systems (23~25 0C). As per the reviewer comments we have added this information in the text (line 87).
Line 168 (Table 3): It will be highly beneficial to the reader if the authors can add the Health status column in Table 3 to Table 1 as it helps the reader understand better the possible contribution of KoRV subtypes to lymphoma.
Response: We thank for the reviewer comments. In agreement with the reviewer comments we have added health status information in updated Table 1.
Line 196: Since these bars in the figure are for individual koalas, kindly explain the error bars shown. Do they refer to the standard deviation of replicates of PBMCs culture (stimulated or unstimulated) or PCR duplicates? Explain this and how many replicates (if PBMC) either in the figure legends or methods section. The same should be done for all figures 1-3.
Response: We thank for the reviewer comments. According to the reviewer comments we have added the error bar information in the revised figure legends (line 203, 226, 243).
Line 219: what do the dashed horizontal lines in the figure indicate? Also, it might be better to present the fold change graphs as a Log2 fold change as such graphs are nicer because they centre around zero. For instance, a fold change of 0.5 would be shown as -1 while a fold change of 2 will be shown as 1 (i.e. fold change = 0.5, Log2FC = -1; fold change=2, Log2FC = 1). Just a suggestion.
Also, for all the bar charts, can the authors attempt to overlay a scatter/dot plot on the bars so the reader can appreciate the individual points being plotted?
Response: We thank for the reviewer comments. To avoid any confusion we have removed the horizontal dashed line. Regarding suggested pattern of the figure, we are making no changes just to keep consistent with other figures style in this manuscript.
Line 227: The authors reported a lower CD4:CD8b ratio in one koala (KM) relative to the other koalas. Did the authors account for seasonal differences as previously reported by Maher et al., 2016 (ref #29)? Was any other koala sampled in similar season? If yes, all good but if not, it might be hard to state that the lowered CD4:CD8 ratio is not a result of seasonal changes in koala immune profile.
Response: We thank the reviewer for his critical comments. The koalas used in this study were maintained in air conditioning systems (23~25 0C) in a Japanese zoo, and therefore no seasonal influence is expected. We have added the housing information of koalas in the revised version of the manuscript (line 87).
Line 267: A previous study also tested immune response in KoRV-A vs KoRVA+B positive koalas in terms of their antibody response to the KoRV antigen (Olagoke et al., 2019; ref #31). I understand that that study did not investigate cytokine or CD4/8 response, but antibody response is also a form of immune response. As such, can the authors rephrase the statement to emphasise the type of immune response they have studied?
Response: We thank the reviewer for his comments. We have rephrased the text by including the immune molecules studied for characterizing immune response (line 272).
Line 278-287: Were these koalas ancestors sourced from Queensland? If yes, that might explain why they behave similarly to northern Australia koala populations re IL17 levels.
Response: We thank the reviewer for his comments. The origins of these populations were not entirely clear. However, among the koalas used in this study, three koalas had been born at a theme park in Queensland.
Line 288: while it is true that the CD4:CD8 ratio can be used as a biomarker for immune evaluation, the truth is that the CD4:CD8 ratio traditionally refers to CD4 and CD8 positive T lymphocytes and which are typically quantified at the cellular and not mRNA level by flow cytometry. To my knowledge, there is no study published study yet on how CD4 and CD8 mRNA levels correlate with actual T lymphocytes counts in koalas. I would argue that you include this point in your interpretation of your really interesting data.
Response: We thank the reviewer for his critical comments. We agree with the reviewer, however in our study we checked the mRNA expression of CD4 and CD8b and calculated the ratio, in a similar pattern of a recently published study (Maher et al., 2016. PLoS One, 5;11(10):e0163780).
The manuscript may benefit from an alignment of the KoRV-A, B and C transmembrane protein p15E to highlight why koalas with exogenous KoRV koalas might be more at risk of immunosuppression. This is just a suggestion and the authors are free to ignore if uninterested.
Response: We thank the reviewer for his comments. In a previous study Ishida et al. 2015 (Virology. 2015 Jan 15; 475: 28–36) demonstrated KoRV p15E is highly conserved among different KoRV subtypes, including those reported to be exogenous. We have added this information (line 304-305).